# A Boundary Offset Prediction Network for Named Entity Recognition

**Minghao Tang**[1,2], **Yongquan He**[3], **Yongxiu Xu**[1,2]*, **Hongbo Xu**[1], **Wenyuan Zhang**[1,2]
and **Yang Lin**[3]

[1]Institute of Information Engineering, CAS, China
[2]School of Cyber Security, UCAS, China
[3]Meituan, China
{tangminghao,xuyongxiu,hbxu}@iie.ac.cn, heyongquan@meituan.com

## Abstract

Named entity recognition (NER) is a fundamental task in natural language processing that aims to identify and classify named entities in text. However, span-based methods for NER typically assign entity types to text spans, resulting in an imbalanced sample space and neglecting the connections between non-entity and entity spans. To address these issues, we propose a novel approach for NER, named the Boundary Offset Prediction Network (BOPN), which predicts the boundary offsets between candidate spans and their nearest entity spans. By leveraging the guiding semantics of boundary offsets, BOPN establishes connections between non-entity and entity spans, enabling non-entity spans to function as additional positive samples for entity detection. Furthermore, our method integrates entity type and span representations to generate type-aware boundary offsets instead of using entity types as detection targets. We conduct experiments on eight widely-used NER datasets, and the results demonstrate that our proposed BOPN outperforms previous state-of-the-art methods.

## 1 Introduction

Named entity recognition (NER) is a fundamental task in natural language processing (NLP) that involves identifying and categorizing named entities in text, such as people, locations and organizations. It has drawn much attention from the community due to its relevance in various NLP applications, such as entity linking (Le and Titov, 2018; Hou et al., 2020) and relation extraction (Miwa and Bansal, 2016; Li et al., 2021b).

Various paradigms have been proposed for NER, including the sequence labeling (Huang et al., 2015; Ju et al., 2018), hypergraph-based (Lu and Roth, 2015; Katiyar and Cardie, 2018; Wang and Lu, 2018), sequence-to-sequence (Gillick et al., 2016; Yan et al., 2021) and span-based methods (Sohrab

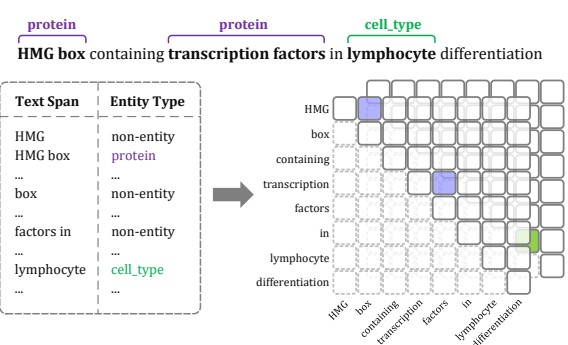

Figure 1: A sentence from GENIA dataset (Ohta et al., 2002), containing 8 words and 3 entities. The candidate spans cover the upper triangular region with a total of 36 samples of each matrix. There are 2 and 1 positive samples for "protein" and "cell type" entity types, respectively.

and Miwa, 2018; Shen et al., 2021; Chen et al., 2021). Among these approaches, the span-based method has become the most popular due to its simplicity and effectiveness. It is straightforward that typically embeds all possible text spans and predicts their entity types, making it suitable for various NER subtasks (Li et al., 2021a, 2022).

Despite significant progress made by span-based methods in NER, there remain two critical issues that require attention. Firstly, these methods often suffer from highly imbalanced sample spaces, as exemplified in Figure 1. Such imbalance can negatively impact the trainability and performance of deep neural networks (Johnson and Khoshgoftaar, 2019). Although some methods (Shen et al., 2021; Wan et al., 2022) mitigate this issue by restricting the maximum span length, such an approach can also constrain the model's predictive power. Secondly, current span-based methods primarily focus on learning the distinction between non-entities and entities, disregarding their relationships. While a model can identify whether "HMG box" is an entity, it may fail to recognize the connection be-

---
*Yongxiu Xu is the corresponding author

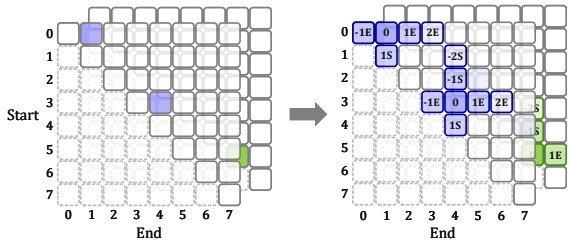

Figure 2: Text spans annotated with boundary offset. "1S" or "1E" represents a span has 1 offset from its nearest entity at the start or end boundary, and so on.

tween "HMG" and "HMG box." To enhance the model's ability to recognize entities, it is crucial to explicitly capture both boundary differences and connections between non-entities and entities.

In this paper, we intend to model text spans by utilizing boundary offset information as supervision, rather than predict their probability of belonging to entities. As shown in Figure 2, there could be two advantages for deep models when boundary offsets are learnable: *i)* The natural quantitative relationships between offset values enable the model to capture boundary differences and connections simultaneously. *ii)* Non-entity spans can have specific semantics that guide the positioning of entity spans, leading to an improved sample space with fewer negative samples.

Based on this observation, we propose the Boundary Offset Prediction Network (BOPN) for NER. BOPN focuses on predicting boundary offsets between candidate spans and their nearest entities, providing a new perspective on modeling text spans. Specifically, our method follows the pipeline of first learning span representations and then classifying them for offset prediction. BERT (Devlin et al., 2019) and BiLSTM (Lample et al., 2016) are used to embed texts, followed by a Conditional Layer (Liu et al., 2021) for building span representations. Meanwhile, we also treat entity types as inputs rather than classification targets, which are fused with span representations to generate type-aware boundary offsets in parallel. Finally, we incorporate multiple 3D convolution layers to capture the natural quantitative relationships between the offset values.

We evaluate our method on eight widely-used NER datasets, including five English NER datasets and three Chinese NER datasets. The experimental results demonstrate that our approach outperforms the existing state-of-the-art methods. Furthermore, a detailed examination reveals a significant im-

provement in recall scores when aggregating results across offset labels, which is particularly beneficial for recall-sensitive applications.

## 2 Problem Definition

Named Entity Recognition (NER) aims to identify of all entities within an input sentence $X = \{x_n\}_{n=1}^N$, based on a pre-defined set of entity types $Y = \{y_m\}_{m=1}^M$. Typically, an entity is specified by token boundaries and a entity types.

Our proposed method focuses on predicting the boundary offset between each candidate text span and its nearest entity. Hence, we formulate each text span as a quadruple: $\{x_i, x_j, f_s, y_m\}$, where $i$ and $j$ denote the start and end boundary indices of the span, $f_s$ represents the start or end boundary offset from its nearest entity of type $y_m$. Note that an entity span is a special case with $f_s = 0$.

**Annotation Guidelines** To facilitate understanding, we present the essential boundary offset labels as follow:

- **Center Span**: refers to an entity span with an offset label of "0".

- **∗S or ∗E**: denotes the annotation of the start or end boundary offsets for non-entity spans. "∗" represents an offset value in the range of $[-S, \cdots, -1, 1, \cdots, S]$, where $S$ denotes the maximum offset value.

- **Out-of-Range**: refers to the annotation of a non-entity span with an absolute boundary offset value from its nearest entity exceeding the maximum offset value $S$.

The annotation procedure for boundary offsets involves three steps. Initially, a 3-dimensional matrix $\mathcal{O} \in \mathbb{R}^{M \times N \times N}$ is constructed according to the input sentence $X$, where $M$ denotes the number of entity types and $N$ represents the length of the sentence. Next, we annotate the center spans with the offset label "0" based on the golden entities present in $X$. Entities of different types are assigned to their respective sub-matrices. Finally, for non-entity spans, we compute the start and end boundary offset values with respect to all center spans. Their annotation is determined by the absolute minimum offset value. If the absolute minimum offset value is less than $S$, we annotate the corresponding ∗S or ∗E; otherwise, we label the span as "Out-of-Range".

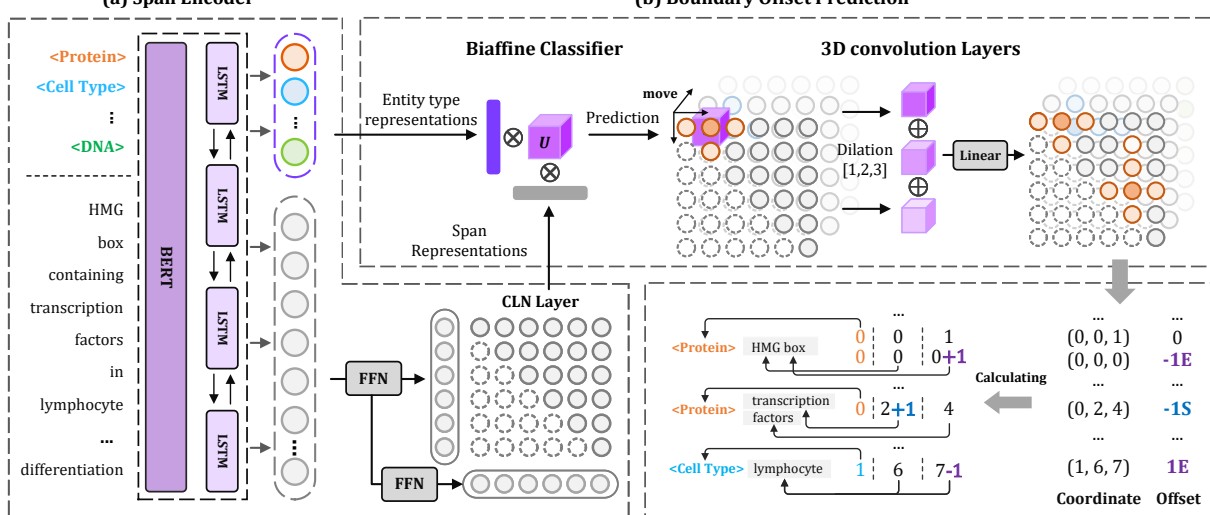

Figure 3: An overview architecture of our method, which mainly consists of two components: a Span Encoder and a Boundary Offset Predictor.

## 3 Methods

Figure 3 provides an overview of our method, which encompasses two primary components: a Span Encoder (Section 3.1) and a Boundary Offset Predictor (Section 3.2). The Span Encoder is responsible for encoding entity types and sentences, utilizing word representations to construct span representations. Subsequently, the entity type and span representations are inputted into the boundary offset predictor, facilitating type-aware offset classification.

### 3.1 Span Encoder

Drawing inspiration from the prompt-based methods (Qin and Eisner, 2021; Han et al., 2022), we consider entity types as task-oriented inputs, indicating the specific types of entities that the model needs to predict within a given sentence.

To achieve this, we create a set of additional type tokens, denoted as $P = \{p_m\}_{m=1}^{M}$, where $p_m$ represents a learnable special token corresponding to entity type $y_m$. Next, we concatenate the soft tokens P with the sentence X to form a single sequence, and employ BERT (Devlin et al., 2019) to encode them simultaneously. The output of BERT is then passed through a BiLSTM (Lample et al., 2016) to generate final embedding features $H = \{h_1, h_2, \cdots, h_{M+N}\} \in \mathbb{R}^{(M+N) \times d}$, where $d$ is the hidden size. Finally, we split H to obtain entity type representations $H^Y \in \mathbb{R}^{M \times d}$ and token representations $H^X \in \mathbb{R}^{N \times d}$, respectively.

**Span Representation**  Given the token representations $H^X = \{h_1, h_2, \cdots, h_N\}$, the span representation $v_{ij}$ can be considered as a fusion of the boundary representations $(h_i, h_j)$. Following Li et al. (2022), we adopt the Conditional Layer Normalization (CLN) (Liu et al., 2021) mechanism to build a high-quality span representation:

$$v_{ij} = \text{CLN}(h_i, h_j)$$
$$= \gamma_j \otimes \text{Norm}(h_i) + \lambda_j, \qquad (1)$$

where $\text{Norm}(\cdot)$ is the instance normalization function (Ulyanov et al., 2016), $\gamma_j$ and $\lambda_j$ are the condition parameters that are obtained by two different feedforward networks: $\gamma_j = \text{FFN}(h_j)$ and $\lambda_j = \text{FFN}(h_j)$.

While valid candidate spans are restricted to the upper triangular region of the adjacent text span matrix, a region embedding $E = [e_{up}, e_{low}] \in \mathbb{R}^{2 \times d_e}$ are adapted to distinguish the positions of text spans. The final representation of each span is obtained as: $\hat{v}_{ij} = [v_{ij}, e_{up}]$ if $i \leq j$; $\hat{v}_{ij} = [v_{ij}, e_{low}]$ if $i > j$.

### 3.2 Boundary Offset Predictor

As previously mentioned, we utilize the entity types as inputs to guide the model in generating type-aware boundary offsets, rather than categorizing each text span into a particular entity type.

The biaffine classifier (Yu et al., 2020) is employed to fuse entity type representations and span representations. Specifically, given an entity type representation $h_m \in H^Y$ and span representation

$\hat{v}_{ij} \in \widehat{V}$, a scoring vector $c_{mij} \in \mathbb{R}^L$ can be computed as:

$$h'_y = \text{FFN}(h_y), \quad \hat{v}'_{ij} = \text{FFN}(\hat{v}_{ij}), \quad (2)$$

$$c_{mij} = (h'_m)^T U \hat{v}'_{ij} + W(h'_m \oplus v'_{ij}) + b, \quad (3)$$

where $L$ is the number of offset labels[1]; $U \in \mathbb{R}^{L \times d_b \times d_b}$, $W \in \mathbb{R}^{L \times 2d_b}$ and $b \in \mathbb{R}^L$ are learnable parameters, $d_b$ is the biaffine hidden size.

**3D Convolution Layer**   Furthermore, we utilize multiple 3-dimensional convolution (3DConv) layers to capture the inherent quantitative relationships between the boundary offsets of adjacent text spans. As depicted in Figure 3(b), the 3D convolution kernels traverse the complete score matrix C in three directions, thereby aggregating offset predictions for adjacent text spans across all entity types. The computation in a single convolution layer can be expressed as:

$$Q = \sigma(3\text{DConv}(C)), \quad (4)$$

where $Q \in \mathbb{R}^{M \times N \times N \times L}$, $\sigma$ is the GELU activation function (Hendrycks and Gimpel, 2016). We assign different dilation rates to each convolution layer, and then concatenate their outputs followed by a linear to calculate final prediction scores:

$$\hat{Q} = \text{Linear}(Q_1 \oplus Q_2 \oplus Q_3), \quad (5)$$

To obtain the probability distribution of span $(i, j)$ over the offset labels, $\hat{q}_{mij} \in \hat{Q}$ is fed into a softmax layer:

$$\hat{o}_{mij} = \text{softmax}(\hat{q}_{mij}), \quad (6)$$

### 3.3   Training and Inference

**Learning Objective**   In our method, the learning objective is to accurately assign a boundary offset to each text span, which can be treated as a multi-class classification problem and optimized using cross-entropy loss:

$$\mathcal{L} = -\frac{1}{MN^2} \sum_m^M \sum_i^N \sum_j^N o_{mij}^T \log(\hat{o}_{mij}) \quad (7)$$

where $o_{mij} \in \mathbb{R}^D$ represents the ground-truth, which is an one-hot vector encoded from the annotated adjacent text span matrix $\mathcal{O}$.

---

[1]Given a maximum offset $S$, $L = 4S + 2$ when considering both start and end boundary offset; $L = 2S + 2$ when only considering start or end boundary offset.

**Inference with Boundary offsets**   During the inference process, decoding entities based on predicted boundary offsets is a straightforward procedure. The output of our method is a matrix of size $M \times N \times N$, where each cell represents a potential entity and contains information about its boundaries and type. For example, a cell with coordinates $(m, i, j)$ and the prediction "-1E" indicates an entity of type $y_m$ with a start boundary at $x_i$ and an end boundary at $x_{j+1}$. Conversely, if the predicted value is "out-of-range," it implies that the cell does not correspond to any entity.

However, blindly accepting all predicted boundary offsets may result in sub-optimal outcomes as it disregards the quantitative relationship between boundary offsets. Therefore, we introduce two heuristic rules to identify unreasonable predictions: *i*) Predicted boundary offsets that do not align with their nearest center span. *ii*) Predicted boundary offsets that do not adhere to a sequential order with neighboring spans.

## 4   Experimental Settings

### 4.1   Datasets

To evaluate our method, we conducted experiments on five English NER datasets, including CoNLL 2003 (Sang and De Meulder, 2003), OntoNotes 5[2], ACE 2004[3], ACE 2005[4] and GENIA (Ohta et al., 2002); and three Chinese NER datasets, including MSRA (Levow, 2006), Resume NER (Zhang and Yang, 2018) and Weibo NER (Peng and Dredze, 2015). Note that ACE 2004, ACE 2005 and GENIA are nested NER datasets, others are flat datasets.

For OntoNotes 5, we take the same train/dev/test as used in CoNLL 2012 shared task (Pradhan et al., 2012). For ACE 2004 and ACE 2005, we use the same data split as Lu and Roth (2015). For GENIA, we follow Katiyar and Cardie (2018) to split train/test as 9:1. For other datasets, we employ the same settings in previous works (Ma et al., 2020; Yan et al., 2021; Zhu and Li, 2022).

### 4.2   Implementation Details

We use BioBERT-v1.1 (Lee et al., 2020) as the contextual embedding in GENIA. For other English corpora, we BERT-large-cased (Devlin et al., 2019) as the contextual embedding. For Chinese

---

[2]https://catalog.ldc.upenn.edu/LDC2005T09
[3]https://catalog.ldc.upenn.edu/LDC2005T09
[4]https://catalog.ldc.upenn.edu/LDC2006T06

| Models | CoNLL 2003 | | | OntoNotes 5 | | |
|---|---|---|---|---|---|---|
| | P | R | $F_1$ | P | R | $F_1$ |
| **Sequence Labeling Methods** | | | | | | |
| BiLSTM-CRF (Miwa and Bansal, 2016) | - | - | 91.03 | 86.04 | 86.53 | 86.28 |
| BERT-Tagger (Devlin et al., 2019) | - | - | 92.80 | 90.01 | 88.35 | 89.16 |
| **Span-based Methods** | | | | | | |
| Biaffine (Yu et al., 2020)*† | 92.46 | 92.67 | 92.55 | 89.94 | 89.81 | 89.88 |
| W2NER (Li et al., 2022) | 92.71 | **93.44** | 93.07 | 90.03 | 90.97 | 90.50 |
| Boundary Smooth (Zhu and Li, 2022)*† | 92.89 | 93.20 | 93.04 | 90.42 | 90.81 | 90.61 |
| DiffusionNER (Shen et al., 2023a) | 92.99 | 92.56 | 92.78 | 90.31 | 91.02 | 90.66 |
| **Others** | | | | | | |
| Seq2Seq (Straková et al., 2019) | - | - | 92.98 | - | - | - |
| BartNER (Yan et al., 2021)† | 92.57 | 93.53 | 93.05 | 89.65 | 90.87 | 90.26 |
| PIQN (Shen et al., 2022) | **93.29** | 92.46 | 92.87 | **91.43** | 90.73 | 90.96 |
| PromptNER (Shen et al., 2023b) | 92.48 | 92.33 | 92.41 | - | - | - |
| **BOPN** (Ours) | 93.22 | 93.15 | **93.19** | 90.93 | **91.40** | **91.16** |

Table 1: Results on English flat NER datasets CoNLL 2003 and OntoNotes 5. † means our re-implementation via their code. * denotes a fair comparison that their BERT encoder is consistent with our model.

| Models | MSRA | | | Resume NER | | | Weibo NER | | |
|---|---|---|---|---|---|---|---|---|---|
| | P | R | $F_1$ | P | R | $F_1$ | P | R | $F_1$ |
| **Sequence Labeling Methods** | | | | | | | | | |
| Lattice (Zhang and Yang, 2018) | 93.57 | 92.79 | 93.18 | 94.81 | 94.11 | 94.46 | 53.04 | 62.25 | 58.79 |
| Flat (Li et al., 2020) | - | - | 96.09 | - | - | 95.86 | - | - | 68.55 |
| SoftLexicon (Ma et al., 2020) | 95.75 | 95.10 | 95.42 | 96.08 | 96.13 | 96.11 | 70.94 | 67.02 | 70.50 |
| MECT (Wu et al., 2021) | - | - | 96.24 | - | - | 95.98 | - | - | 70.43 |
| **Span-based Methods** | | | | | | | | | |
| W2NER (Li et al., 2022) | 96.12 | 96.08 | 96.10 | 96.96 | 96.35 | 96.65 | 70.84 | 73.87 | 72.32 |
| Boundary Smooth (Zhu and Li, 2022) | 96.37 | 96.15 | 96.26 | 96.63 | 96.69 | 96.66 | 70.16 | **75.36** | 72.66 |
| DiffusionNER (Shen et al., 2023a) | 95.71 | 94.11 | 94.91 | - | - | - | - | - | - |
| **BOPN** (Ours) | **96.44** | **96.34** | **96.39** | **96.73** | **96.83** | **96.78** | **71.79** | 73.90 | **72.92** |

Table 2: Results on Chinese flat NER datasets MSRA, Resume and Weibo.

corpora, we use the BERT pre-trained with whole word masking (Cui et al., 2021).

The BiLSTM has one layer and 256 hidden size with dropout rate of 0.5. The size of region embedding $d_e$ is 20. The maximum offset value $S$ is selected in $\{1, 2, 3\}$. For all datasets, we train our models by using AdamW Optimizer (Loshchilov and Hutter, 2017) with a linear warmup-decay learning rate schedule. See Appendix A for more details. Our source code can be obtained from https://github.com/mhtang1995/BOPN.

## 4.3 Evaluation

We use strict evaluation metrics where a predicted entity is considered correct only when both the boundaries (after adding boundary offset) and type are accurately matched. The precision, recall and $F_1$ scores are employed. We run our model for five times and report averaged values.

## 5 Results and Analysis

### 5.1 Main Results

The performance of our proposed method and the baselines on English flat NER datasets is presented in Table 1. The experimental results demonstrate that our approach surpasses the previous state-of-the-art (SOTA) methods by +0.12% on the CoNLL 2003 dataset and +0.20% on the OntoNotes 5 dataset, achieving superior performance with $F_1$ scores of 93.19% and 91.16%, respectively. For Chinese flat NER datasets, we provide the results in Table 2. Similarly, our proposed method achieves SOTA performance in terms of $F_1$ scores, surpassing the previous best method by +0.13%, +0.12%, and +0.26% in $F_1$ scores on the MSRA, Resume NER, and Weibo NER datasets, respectively.

The performance results on English nested NER datasets are presented in Table 3. Remarkably,

| Models | ACE 2004 | | | ACE 2005 | | | GENIA | | |
|---|---|---|---|---|---|---|---|---|---|
| | P | R | $F_1$ | P | R | $F_1$ | P | R | $F_1$ |
| **Sequence Labeling Methods** | | | | | | | | | |
| Layered (Ju et al., 2018) | - | - | - | 74.2 | 70.3 | 72.2 | 78.5 | 71.3 | 74.7 |
| Pyramid (Wang et al., 2020) | 86.08 | 86.48 | 86.28 | 83.95 | 85.39 | 84.66 | 79.45 | 78.94 | 79.19 |
| **Span-based Methods** | | | | | | | | | |
| Biaffine (Yu et al., 2020) | 87.3 | 86.0 | 86.7 | 85.2 | 85.6 | 85.4 | 78.2 | 78.2 | 78.2 |
| Locate and Label (Shen et al., 2021) | 87.44 | 87.38 | 87.41 | 86.09 | 87.27 | 86.67 | 80.19 | 80.89 | 80.54 |
| W2NER (Li et al., 2022) | 87.33 | 87.71 | 87.52 | 85.03 | 88.62 | 86.79 | 83.10 | 79.76 | 81.39 |
| Triaffine (Yuan et al., 2022) | 87.13 | 87.68 | 87.60 | 86.70 | 86.94 | 86.82 | 80.42 | **82.06** | 81.23 |
| Boundary Smooth (Zhu and Li, 2022) | 88.43 | 87.53 | 87.98 | 86.25 | 88.07 | 87.15 | - | - | - |
| DiffusionNER (Shen et al., 2023a) | 88.11 | 88.66 | 88.39 | 86.15 | 87.72 | 86.93 | 82.10 | 80.97 | 81.53 |
| **Others** | | | | | | | | | |
| Seq2Seq (Straková et al., 2019) | - | - | 84.33 | - | - | 83.42 | - | - | 78.20 |
| BartNER (Yan et al., 2021) | 87.27 | 86.41 | 86.84 | 83.16 | 86.38 | 84.74 | 78.57 | 79.30 | 78.93 |
| PIQN (Shen et al., 2022) | 88.48 | 87.81 | 88.14 | 86.27 | 88.60 | 87.42 | **83.24** | 80.35 | 81.77 |
| PromptNER (Shen et al., 2023b) | 87.58 | 88.76 | 88.16 | 86.07 | 88.38 | 87.21 | - | - | - |
| **BOPN** (Ours) | **89.13** | **89.40** | **89.26** | **89.56** | **91.23** | **90.39** | 82.14 | 82.16 | **82.14** |

Table 3: Results on English nested NER datasets ACE 2004, ACE 2004 and GENIA.

| | CoNLL 2003 | Resume NER | ACE 2004 |
|---|---|---|---|
| **BOPN** (Ours) | **93.19** | **96.78** | **89.26** |
| - w/o Type Inp. | 92.87 | 96.41 | 88.83 |
| - w/o Region Emb. | 92.71 | 96.22 | 88.71 |
| - w/o BO | 92.74 | 96.26 | 88.62 |
| - w/o 3DConv | 92.87 | 96.40 | 89.11 |
| - MBO ($S = 1$) | 93.11 | 96.75 | 89.14 |
| - MBO ($S = 2$) | 93.15 | **96.78** | **89.26** |
| - MBO ($S = 3$) | **93.19** | 96.71 | 89.22 |
| - 3DConv ($l = 1$) | 93.08 | 96.69 | 89.18 |
| - 3DConv ($l = 2$) | **93.19** | 96.75 | **89.26** |
| - 3DConv ($l = 3$) | 93.05 | **96.78** | 89.25 |

Table 4: Ablation Studies. MBO means the maximum boundary offset value.

our proposed BOPN achieves substantial improvements in performance on these datasets, with $F_1$ scores increasing by +0.87%, +2.97%, and +0.37% on ACE 2004, ACE 2005, and GENIA, respectively. These results align with our expectations, as the boundary features of nested entities are more intricate compared to flat entities. We attribute this improvement to two key factors: 1) Our method predicts the boundary information of various entity types in parallel, effectively avoiding nested boundary conflicts between different types of entities. 2) By predicting boundary offsets, our method expands the predictive range for each text span, allowing for more granular and precise identification of entity boundaries.

## 5.2 Ablation Studies

In order to assess the impact of each component in our method, we conduct ablation studies on the CoNLL 2003, ACE 2005, and Resume NER datasets. The results of these studies are presented in Table 4.

**Maximum Boundary Offset** We investigate the impact of training the model with different maximum offset values $S$ through our ablation studies. The hyperparameter $S$ determines the annotation scope of non-entity spans with boundary offset. Specifically, the extreme scenario of setting $S$ to 0 corresponds to a condition "w/o BO" (without Boundary Offset). The results indicate a significant decline in performance when employing "w/o BO," confirming the usefulness of utilizing boundary offsets as supervision. However, we also observe that the optimal $S$ value varies across different datasets. This could be attributed to the fact that a larger $S$ value provides more boundary knowledge but also increases the label search space. Consequently, hyperparameter tuning for $S$ becomes necessary to achieve the best performance in practice.

In addition, we analyze the learning curves of our model with different maximum offset values. Figure 4 demonstrates that a larger $S$ can accelerate the training process of the model. We think the reason may be that a larger $S$ not only leads to an increase of positive samples but also results in a decrease of negative samples, thereby ultimately enhancing the trainability of the model.

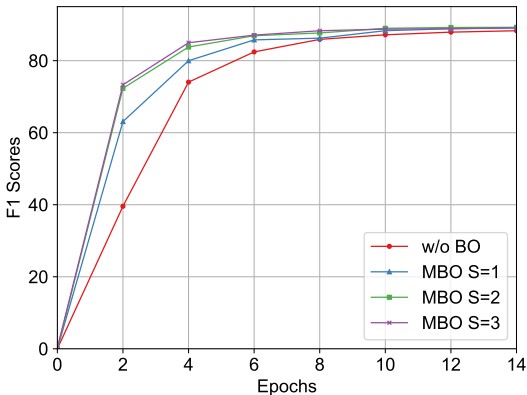

Figure 4: The learning curves on ACE 2004 dataset.

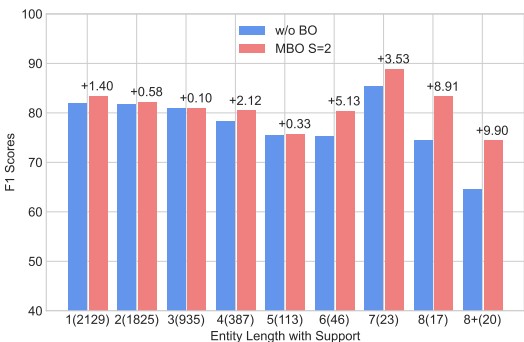

Figure 5: A comparison of F1-scores on entities of different lengths in GENIA dataset. Entity supports are in the parentheses.

| Label | P | R | $F_1$ | Support |
|---|---|---|---|---|
| -2S | 81.51 | 82.02 | 81.76 | 5029 |
| -1S | 81.62 | 82.97 | 82.29 | 5292 |
| 1S | 79.55 | 81.47 | 80.50 | 3281 |
| 2S | 76.27 | 79.55 | 77.88 | 1438 |
| -2E | 78.64 | 77.19 | 77.90 | 1464 |
| -1E | 79.79 | 80.58 | 80.18 | 3254 |
| 1E | 82.26 | 82.20 | 82.23 | 5393 |
| 2E | 82.37 | 80.75 | 81.57 | 5113 |
| 0 | **81.92** | 81.95 | 81.93 | **5495** |
| ALL | 79.21 | **84.22** | 81.64 | 5495 |
| - w/ rules | 81.85 | 82.56 | **82.20** | 5495 |

Table 5: Performance of each boundary offset label on GENIA, where the maximum offset value is 2. The reported results is one out of five experiments.

**3D Convolution Layer**  "w/o 3DConv" indicates the 3D convolution layers are removed. As seen, the results show a decline in performance across all datasets, indicating the importance of 3D convolution layers in capturing the interactions between boundary offsets of adjacent text spans.

**Type Inputs**  "w/o Type Inputs" refers to a setting where the entity types encoded with the sentence are replaced, in which the randomly initialized entity type embeddings are fed into the biaffine classifier. The results obtained in this setting show a slight decline in performance.

**Region Embedding**  The results demonstrate a slight drop in performance across all datasets without region embeddings. This suggests that integrating sample distribution features can be a reasonable approach for enhancing text span representations.

As the CLN layer and biaffine classifier serve as fundamental components in our approach for span representation and classification, they cannot be evaluated independently. Nonetheless, our ablation studies demonstrate the effectiveness of learning boundary offset information and the usefulness of each composition in our model.

### 5.3 Detailed Analysis

**Performance on Different Offset Labels**  We investigate the performance of each boundary offset label, and the results are presented in Table 5. Notably, the offset label "0" has complete entity support and achieves an $F_1$ score of 82.04%. Furthermore, we observed a positive correlation between the quantity of entity support and the performance of boundary offset labels.

When a text span is not predicted as "out-of-range", its assigned label can be utilized to determine the position of its nearest entity. By aggregating all predictions of offset labels, we observe a sharp decrease in precision score, along with a significant increase in recall score, when compared to only considering the center span (with an offset label of "0"). This finding suggests that different offset labels provide distinct information that assists the model in recognizing additional entities. Nevertheless, this approach can introduce noisy predictions due to the model's inadequate performance on certain labels. Despite this limitation, it may have practical applicability in recall-sensitive applications.

As discussed in Section 3.3, we devise two heuristic rules to remove improbable predictions. Our findings reveal that this approach enhances the precision score, with only a minor reduction in the recall score, leading to an overall improvement in the $F_1$ score.

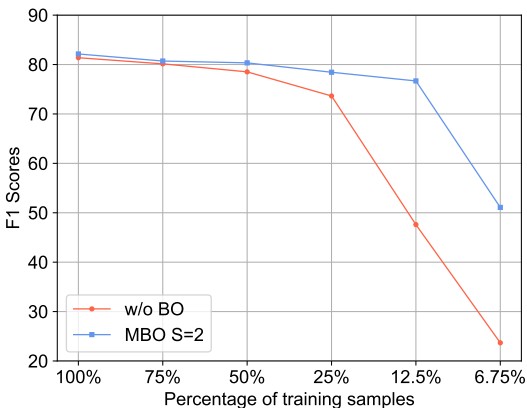

Figure 6: Effect of varying percentage of training samples on GENIA. We train all models for 50 epochs and report their best performance.

**Performance on Entities with Varying Lengths**
We explore the model performance on entities of different lengths in GENIA. As shown in Figure 5, we compare the $F_1$ scores of models which are training with different $S$. The model achieves higher $F_1$ scores across all columns when $S = 2$, with a more pronounced performance improvement for longer entities. The results highlight the usefulness of learning boundary offsets between non-entity and entity spans, which helps the model learn boundary features more effectively.

**Size of Training Data** As the boundary offset labels contain more informative knowledge, we hypothesize that our proposed BOPN would perform better with limited training data. As shown in Figure 6, our model achieves impressive results, exhibiting only a 5.46% decrease in performance when trained with a mere 12.5% of the available training data. In contrast, when boundary information is not utilized during training, the model's performance declines rapidly as the amount of training data decreases, thus creating significant obstacles to effective training.

## 6 Related Work

In recent years, various paradigms for named entity recognition (NER) have been proposed, among which span-based methods have become one of the most mainstream approaches, treating NER as a text span classification problem. With the development of pre-trained language models, some works (Sohrab and Miwa, 2018; Luan et al., 2019; Wadden et al., 2019) obtain span representations by connecting boundary representations or aggregating token representations and feeding them into a linear classifier for type prediction. Alternatively, Yu et al. (2020) utilizes a biaffine classifier to fuse start and end boundary representations directly for span classification. To further enhance span representation, several other methods(Wan et al., 2022; Yuan et al., 2022) propose fusing representations of token, boundary, and related entity spans.

Meanwhile, some methods try to improve span-based methods by adding boundary supervision. Specifically, Zheng et al. (2019) and Tan et al. (2020) additionally detect entity boundaries with multi-task learning, while Shen et al. (2021) perform boundary regression after span prediction. Li et al. (2022) design two word-word relations for span classification. Compared with previous methods, our proposed method utilizes continuous boundary offset values to model text spans, which can capture both the boundary differences and connections between non-entity and entity spans.

In addition to span-based methods, there are three widely-used NER methods. The traditional sequence labeling methods (Huang et al., 2015; Lample et al., 2016) assign each token a tag with a pre-designed tagging scheme (e.g., *BIO*). To address nested entities, some works (Ju et al., 2018; Wang et al., 2020; Rojas et al., 2022) add struggles or design special tagging schemes. Hypergraph-based methods (Lu and Roth, 2015; Katiyar and Cardie, 2018; Wang and Lu, 2018) represent the input sentence as a hypergraph for detecting nested entities, which must be carefully designed to avoid spurious structures. Sequence-to-sequence methods reformulate NER as a sequence generation problem. For example, Gillick et al. (2016) first apply the Seq2Seq model for NER, inputting the sentence and outputting start positions, entity lengths, and types. Straková et al. (2019) use the Seq2Seq model and enhanced BILOU scheme to address nested NER. Yan et al. (2021) treats NER as an entity span sequence generation problem with pointer network based on BART (Lewis et al., 2019).

## 7 Conclusion

In this paper, we introduce a novel approach for named entity recognition (NER) called the Boundary Offset Prediction Network (BOPN). BOPN predicts the boundary offsets between candidate spans and their nearest entities, leveraging entity types as inputs. By incorporating entity types, BOPN enables parallel prediction of type-aware boundary offsets, enhancing the model's ability to capture

fine-grained entity boundaries. To capture the interactions between boundary offsets, we employ multiple 3D convolution layers, which refine the offset predictions and capture the inherent quantitative relationships between adjacent text spans.

The experimental results demonstrate that our proposed method achieves state-of-the-art performance on eight widely-used datasets, including five English NER datasets and three Chinese NER datasets. Moreover, further analysis reveals a significant improvement in recall scores by utilizing boundary offset as supervision, showcasing the utility of our approach for recall-sensitive applications in NER.

## Limitations

The proposed BOPN approach has certain limitations that should be acknowledged. Firstly, while BOPN treats boundary offsets as classification targets, it does not explicitly model the order relationship between offset values. Although the 3D convolution layers are employed to implicitly capture interactions between boundary offsets, they do not provide a strong constraint on the ordering of offset labels.

Additionally, the method uses boundary offsets to convert some non-entity spans into positive samples, which leads to higher recall scores but potentially lower precision scores. To optimize prediction results, heuristic rules are applied to filter out unreasonable samples. However, these rules are based on observations and may not be comprehensive enough to handle all cases effectively.

Therefore, there is still a need to explore more effective ways to integrate and optimize the offset predictions in order to address these limitations and enhance the overall performance of the BOPN approach.

## Ethics Statement

To address ethical concerns, we provide the two detailed description: 1) All experiments were conducted on existing datasets derived from public scientific papers. 2) Our work does not contain any personally identifiable information and does not harm anyone.

## Acknowledgements

This work was supported by Strategic Priority Research Program of Chinese Academy of Sciences (N0. XDC02040400).

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

## A  Appendix

### A.1  Datasets

We evaluate our method on eight datasets, including CoNLL 2003, OntoNotes 5, ACE 2004, ACE 2005, and GENIA for English NER datasets; MSRA, Resume NER and Weibo NER for Chinese NER datasets. Table 6 presents the detailed statistics of datasets.

### A.2  Implementation Details

We use BioBERT-v1.1 (Lee et al., 2020) as the contextual embedding in GENIA. For other English corpora, we BERT-large-cased (Devlin et al., 2019) as the contextual embedding. For Chinese corpora, we use the BERT pre-trained with whole word masking (Cui et al., 2021). Our model is implemented with PyTorch and trained with a NVIDIA RTX3090 GPU. We use a grid search to find the best hyperparameters which are tuned on the development set. The range of hyperparameters we used for eight datasets are listed in Table 7.

### A.3  Baselines

We compare BOPN with the following baselines:

- **BiLSTM-CRF** (Miwa and Bansal, 2016) is a model for sequence labeling tasks that combines BiLSTM with CRF layers.

- **BERT-Tagger** (Devlin et al., 2019) that utilizes the pre-trained language model BERT as a feature extractor and incorporates a tag classifier for fine-tuning.

- **Lattice** (Zhang and Yang, 2018) proposed a lattice-structured LSTM model for Chinese NER.

- **Layered** (Ju et al., 2018) dynamically stacks flat NER layers to solve nested NER task.

- **Flat** (Li et al., 2020) proposes a flat-lattice transformer for Chinese NER, which converts the lattice structure into a flat structure consisting of spans.

- **Pyramid** (Wang et al., 2020) designs pyramid layer and inverse pyramid layer to decode nested entities.

- **SoftLexicon** (Ma et al., 2020) proposes a Chinese NER method in which lexicon information is introduced by simply adjusting the character representation layer.

- **MECT** (Wu et al., 2021) uses multi-metadata embedding in a two-stream transformer to integrate Chinese character features with the radical-level embedding.

- **Biaffine** (Yu et al., 2020) classifies text spans by a biaffine classifier between boundary representations.

- **Locate and Label** (Shen et al., 2021) proposed a two-stage identifier of locating entities with boundary regression first and classifying them later.

- **W2NER** (Li et al., 2022) models NER as word-word relation classification, including the next-neighboring-word and the tail-head-word relations.

- **Triaffine** (Yuan et al., 2022) proposed a tri-affine mechanism to fuse information of inside tokens, boundaries, labels for NER.

- **Boundary Smooth** (Zhu and Li, 2022) proposed boundary smoothing as a regularization technique for span-based neural NER models.

- **DiffusionNER** (Shen et al., 2023a) formulates NER as a boundary-denoising diffusion process, which samples noisy spans from a Gaussian distribution.

- **Seq2Seq** (Straková et al., 2019) converts the labels of nested entities into a sequence and then uses a seq2seq model to decode entities.

- **BartNER** (Yan et al., 2021) formulates NER as an entity span sequence generation problem based on the pre-training Seq2Seq model BART (Lewis et al., 2019).

- **PIQN** (Shen et al., 2022) sets up global and learnable instance queries to extract entities from a sentence in a parallel manner.

- **PromptNER** (Shen et al., 2023b) unifies entity locating and entity typing in prompt learning for NER, which predicts all entities by filling position slots and type slots.

| | CoNLL 2003 | OntoNotes 5 | ACE 2004 | ACE 2005 | GENIA | MSRA | Resume | Weibo |
|---|---|---|---|---|---|---|---|---|
| Types | 4 | 18 | 7 | 7 | 5 | 3 | 8 | 4 |
| #Train.S | 17291 | 59924 | 6200 | 7194 | 16692 | 46471 | 3819 | 1350 |
| #Dev.S | - | 8528 | 745 | 969 | - | - | 463 | 270 |
| #Test.S | 3453 | 8262 | 812 | 1047 | 1854 | 4376 | 477 | 270 |
| Avg.Len.S | 14.38 | 18.11 | 22.61 | 18.97 | 25.41 | 45.54 | 31.17 | 54.57 |
| #Train.E | 29441 | 128738 | 22204 | 9389 | 50509 | 74703 | 13438 | 1855 |
| #Dev.E | - | 20354 | 2514 | 1112 | - | - | 1497 | 379 |
| #Test.E | 5648 | 12586 | 3035 | 1118 | 5506 | 6181 | 1630 | 409 |
| Avg.Len.E | 1.45 | 1.83 | 2.50 | 2.28 | 1.97 | 3.24 | 5.88 | 2.60 |

Table 6: Dataset Statistics. "#" denotes the amount. "S." and "E." denote sentence and entity mentions, respectively.

| Parameter | Value |
|---|---|
| Epoch | [50, 80] |
| Batch size | [8, 16] |
| Learning rate (BERT) | [5e-6, 3e-5] |
| Learning rate (Other) | 1e-3 |
| LSTM hidden size $d$ | 256 |
| LSTM dropout | 0.5 |
| Region embedding size $d_e$ | 20 |
| Biaffine hidden size $d_b$ | 150 |
| Biaffine dropout | 0.2 |
| Maximum offset value $S$ | [1, 3] |
| Adam epsilon | 1e-8 |
| Warm factor | 0.1 |

Table 7: Hyper-parameter settings.