# OpenReview forum: "A Boundary Offset Prediction Network for Named Entity Recognition"
_EMNLP/2023/Conference — EMNLP 2023 Findings_

### Official Review · Reviewer_TfVb · 2023-08-02

**Soundness:** 3

**Excitement:**

2: Mediocre: This paper makes marginal contributions (vs non-contemporaneous work), so I would rather not see it in the conference.

**Paper Topic And Main Contributions:**


Boundary Offset Prediction Network (BOPN) is proposed to predict the boundary offsets between candidate spans and their nearest entity spans. Experiments are conducted on eight widely-used NER datasets.


**Questions For The Authors:**

What are the pros and cons compared to recent generative IE models?

**Reasons To Accept:**

1. Extensive experiments on eight datasets.
2. Good technical presentation.


**Reasons To Reject:**

1.The authors claim that BOPN achieves state-of-the-art performance. However, it is still far from the real SOTA. e.g.,
 2021 Automated Concatenation of Embeddings for Structured Prediction. CoNLL 03 F1: 94.6
2019 Dice Loss for Data-imbalanced NLP Tasks. Ontonotes F1 92.07

2. Some recent generative models are not included.


**Reproducibility:**

3: Could reproduce the results with some difficulty. The settings of parameters are underspecified or subjectively determined; the training/evaluation data are not widely available.

**Reviewer Confidence:**

4: Quite sure. I tried to check the important points carefully. It's unlikely, though conceivable, that I missed something that should affect my ratings.

---

> ### Author Rebuttal · Authors · 2023-08-28
>
> Thanks for the valuable comments. Here are our replies:
>
> Q1：About the comparison with the paper [1] ("Automated Concatenation of Embeddings for Structured Prediction") on CoNLL 2003 dataset.
>
>
> A1: We did not chosen paper[1] as our baseline model, mainly for the following two reasons:
>
>
> 1) The research motivations differ significantly. While our method focuses on innovating the NER model architecture, the primary objective of the paper [1] is to find better word representations rather than improving model architectures.
>
>
> 2) Comparing our method with the paper [1] would introduce unfairness  due to differences in experimental settings. The paper [1] proposes an algorithm that automatically combines 11 auxiliary resources to generate word embeddings. In contrast, our approach emphasizes innovation on model architecture and solely uses BERT-large for word embeddings. In Table 1 of the paper [1], the combination of nine resources (ELMo, Flair, base BERT, GloVe word embeddings, fastText word embeddings, non-contextual character embeddings, multilingual Flair (M-Flair), M-BERT, and XLM-R embeddings) achieves 93.0 F1 on CoNLL 2003. However, our method achieves a slightly higher F1 of 93.19 using only BERT-large.  In Table 2 of the paper [1], they further add XLNet and RoBERTa in addition to the nine resources, resulting in an F1 score of 94.6. It is known that XLNet and RoBERTa can  produce more powerful word embeddings. We would like to share  experimental results that replaces BERT-large with XLNet and RoBERTa in our method, which achieves F1 scores of 94.5 and 94.2 F1 on CoNLL 2003, respectively.
>
>
> We genuinely hope that our responses address your queries and contribute to a positive outcome. It is crucial to ensure fair conditions for comparisons. We appreciate your understanding and continued support.
>
>
> Q2: About the comparison with the paper [2] ("Dice Loss for Data-imbalanced NLP Tasks") on Ontonotes dataset.
>
>
> A2: Similarly, we refrain from comparing our approach with paper [2] due to the different motivations. Paper [2] proposes the dice-based loss function to enhance model performance, while our method aims to innovate the NER model architecture. Paper [2] utilizes the MRC-NER model as the backbone. In a fair environment with the same loss function and experimental settings, our method demonstrates comparable performance on Ontonotes dataset, achieving 91.16 (ours) vs. 91.11 (MRC-NER).
>
>
> Q3：About the pros and cons compared to recent generative IE models.
>
>
> A3: We have referenced two influential papers, [3] (A unified generative framework for various NER subtasks, 2021) and [4] (GPT-NER: Named entity recognition via large language models, 2023). The pros and cons of our approach are as follows:
>
> Pros:
>
>
> Personalize: Generic generative IE models commonly aim to handle IE tasks with a unified structure, in which providing personalize service would be challenging. Our method leverages entity-relative positional information to model entity spans, which can capturing both the differences and connections between entity and non-entity spans. Moreover, by incorporating predictions of all boundary offsets, our approach provides more comprehensive entity predictions, significantly improving recall performance. This is a capability not possessed by generic generative IE models.
>
>
> Inference Speed: Our method utilizes BERT as the context embedding layer, which has a time complexity of O(n^2). In contrast, generative models typically are decode-only models with a time complexity of O(n^3), even with optimizations like FlashAttention and Multi-query Attention. Inference speed is crucial for NER as it constitutes a fundamental step in information extraction.
>
>
> Performance: In the fully supervised setting, our method achieves higher performance compared to paper [3] and [4] under relatively fair conditions.
>
>
> Accessibility: Considering hardware resources (GPU) and training time costs, our method may be more easily accessible. GPT-3.5 requires the use of an API, while other open-source generative models (e.g., T5, llama) require more GPU resources.
>
> Cons:
>
>
> Few-shot Capability: Paper [4] demonstrates exceptional performance in few-shot and zero-shot scenarios, an area where our method may be comparatively weaker.
>
>
> Longer Sentences: Paper [4] can process up to 4k tokens in a single instance.
>
> We genuinely appreciate your feedback and suggestions, and we will incorporate them into our work to improve its quality. We highly appreciate your time, effort, and consideration.
>
>
> [1] Automated Concatenation of Embeddings for Structured Prediction (2021)
>
>
> [2] Dice Loss for Data-imbalanced NLP Tasks (2020)
>
>
> [3] A unified generative framework for various NER subtasks (2021)
>
>
> [4] GPT-NER: Named entity recognition via large language models (2023)

---

### Official Review · Reviewer_no4M · 2023-08-03

**Soundness:** 4

**Excitement:**

4: Strong: This paper deepens the understanding of some phenomenon or lowers the barriers to an existing research direction.

**Paper Topic And Main Contributions:**

The paper introduces a new approach for the NER task based on the idea that the relationships between entity and non-entity text spans matter, and accordingly, the boundary offsets of candidate spans relative to the nearest entity span helps improve the prediction. The authors design a boundary offset prediction network (BOPN) that, unlike conventional span-based NER modeling, model the problem as a multi-class classification task where the output space involves predicting the offset labels, corresponding to whether a span refers an entity span, is away from the nearest entity span by some "allowable" offset, or is outside the maximum offset value. Another flexibility of the model is that it is designed to be type-aware by including the entity type as an input (instead of output), with a trainable token. The approach is aimed to tackle two challenges with span-based deep networks performing NER: 1) addressing the imbalance problem between entity and non-entity spans by leveraging their relative positioning, and 2) not ignoring the relationship between entity and non-entity span, but instead using the non-entity spans to provide better positive signals. As such, the approach not only fares better than state-of-the-art methods, but is also shown to improve recall.

The paper starts with a good brief introduction to the problem, drawing the reader gradually to the two key challenges that motivate the new design. The intuition behind the design, the architecture, and all modeling aspects (annotation guidelines, representation, prediction network, training, inference) have been explained well, with diagrams, examples and appropriate level of equations and citations that convey all the important details. The results of the approach are clear from the extensive experiments performed on 8 datasets (5 English, 3 Chinese).

While the results outperform existing methods, they are far from perfect. Therefore, the paper could be improved by including a discussion of when the technique falls short. Please note, this is different from the limitations of the approach, in that it is about supporting results with error analysis that help readers understand patterns that cannot be addressed or continue to remain challenging.

**Reasons To Accept:**

A. The paper is clear in its motivation, problem formulation, overall description of the method and results.

B. The paper improves the NER performance by modeling it a bit differently than ongoing span-based models for NER and leveraging effectively the relative positioning information between entity and non-entity spans. The potential to boost recall can be particularly attractive for practitioners.

C. The authors perform extensive experiments on 8 datasets from 2 languages and with varying complexity (number of entity types, presence of nested entities, etc.), not only convincing the reader of the overall results, but also clearly demonstrating the effects of various architectural choices, effects of text span sizes and training parameter choices.

**Reasons To Reject:**

A. The paper readability can be improved with a round of proof-reading, in that certain typos can be fixed.

B. While the Limitation section introspects the shortcomings of the approach, the paper can be improved by discussing the major error patterns that still remain despite outperforming SOTA methods.

**Reproducibility:**

4: Could mostly reproduce the results, but there may be some variation because of sample variance or minor variations in their interpretation of the protocol or method.

**Reviewer Confidence:**

4: Quite sure. I tried to check the important points carefully. It's unlikely, though conceivable, that I missed something that should affect my ratings.

**Typos Grammar Style And Presentation Improvements:**

- L124: Omit 'that'
- Fig 3: 'The' -> 'An'
- L193: 'an region embedding' -> 'a region embedding'
- L351: 'Thel' -> 'The'
- L355: Omit redundant 'datasets'

---

> ### Author Rebuttal · Authors · 2023-08-28
>
> Thank you so much for your invaluable and encouraging feedback. We deeply appreciate your thoughtful comments, and we are grateful for the opportunity to incorporate your suggestions. Your insights will undoubtedly help us improve the readability of our paper and provide a more comprehensive discussion about the major error patterns in the final version of our work. Once again, thank you for your invaluable contribution.

---

### Official Review · Reviewer_GY7H · 2023-08-06

**Soundness:** 3

**Excitement:**

3: Ambivalent: It has merits (e.g., it reports state-of-the-art results, the idea is nice), but there are key weaknesses (e.g., it describes incremental work), and it can significantly benefit from another round of revision. However, I won't object to accepting it if my co-reviewers champion it.

**Paper Topic And Main Contributions:**

This paper proposes a new span-based NER model that solves the task by predicting boundary offsets of named entities. The main idea is that it uses an offset to the nearest entity span as the target labels of each candidate span. Furthermore, it assigns input embeddings to entity types and treats them as inputs. These techniques enhance the expressive power of the model, improving the NER performance.

The paper is well-written and easy to read. The model's effectiveness is tested using extensive experiments on multiple popular NER datasets.

One weakness is that, although the proposed model is relatively complex, the performance improvement compared to the existing models is marginal. For example, without using the 3D convolution layer, the model does not outperform the previous models on CoNLL 2003 and Resume NER.

**Reasons To Accept:**

* The paper proposes a new span-based NER model with state-of-art performance
* Extensive experiments are conducted on multiple popular NER datasets
* The paper is well-written

**Reasons To Reject:**

* The improvement of the NER performance is relatively marginal

**Reproducibility:**

4: Could mostly reproduce the results, but there may be some variation because of sample variance or minor variations in their interpretation of the protocol or method.

**Reviewer Confidence:**

3: Pretty sure, but there's a chance I missed something. Although I have a good feel for this area in general, I did not carefully check the paper's details, e.g., the math, experimental design, or novelty.

---

> ### Author Rebuttal · Authors · 2023-08-28
>
> Thank you so much for your valuable comments. We truly appreciate your comments. Here are our responses:
>
> Q: The improvement of the NER performance is relatively marginal (compared to the existing models).
>
> A: We acknowledge that the existing NER methods have made significant advancements in recent years, reaching higher performance levels and achieving substantial gains has become increasingly challenging. Different from previous methods, our approach mainly focuses on  optimizing imbalanced sample spaces by considering the entity-relative position information. Specifically, we employ multiple 3D convolution layers to explicitly capture the the inherent quantitative relationships between the boundary offsets of adjacent text spans. However, it is important to recognize that every incremental performance improvement requires great effort. For instance, the previous state-of-the-art model (W2NER 2022) achieved a 93.07% F1 score on CoNLL 2003, only surpasses the baseline model (Bert-Tagger) by 0.27% F1 score.While our method outperforms Bert-Tagger by 0.39% F1 on CoNLL 2003.
>
>
> We also beg to provide clear clarification that we conducted experiments on eight widely-used NER datasets, and our method consistently demonstrated superior performance across all of them. Specifically, we achieved boosted F1 scores of +0.87%, +2.97%, and +0.37% on ACE 2004, ACE 2005, and GENIA, respectively. Moreover, our method significantly enhances recall performance by effectively aggregating all boundary offset results. This innovative approach offers practitioners new insights, particularly in industries where recall holds greater importance than F1 performance.
>
>
> We genuinely hope that our responses address your queries and contribute to a positive outcome. Your time, effort, and consideration are highly valued and deeply appreciated.

---

### Meta-Review · Area_Chair_rTzh · 2023-09-19

**Recommendation:** 3

**Metareview:**

While the reviews have a consensus on positive soundness, there is a discrepancy in the excitement evaluation.

Regarding soundness, the paper conducts extensive experiments on 8 datasets from 2 languages, supporting the effectiveness of the proposed method.  The paper would be more insightful if it provided an error analysis.  All the reviewers see the paper is well-written.

As for excitement, the proposed span-based NER model is novel.  The result of improving recall has a practical impact.  I would lean toward the slightly positive side in excitement, mainly because of how the proposed method addresses two problems with existing span-based NER models.

---

### Decision · Program_Chairs · 2023-10-07

**Decision:**

Accept-Findings

**Comment:**

While the reviews have a consensus on positive soundness, there is a discrepancy in the excitement evaluation.

Regarding soundness, the paper conducts extensive experiments on 8 datasets from 2 languages, supporting the effectiveness of the proposed method.  The paper would be more insightful if it provided an error analysis.  All the reviewers see the paper is well-written.

As for excitement, the proposed span-based NER model is novel.  The result of improving recall has a practical impact.  I would lean toward the slightly positive side in excitement, mainly because of how the proposed method addresses two problems with existing span-based NER models.